# Recent Advances in the Structural Studies of the Proteolytic ClpP/ClpX Molecular Machine

**DOI:** 10.3390/biom15081097

**Published:** 2025-07-29

**Authors:** Astrid Audibert, Jerome Boisbouvier, Annelise Vermot

**Affiliations:** Univ. Grenoble Alpes, CNRS, CEA, Institut de Biologie Structurale (IBS), 71, Avenue des Martyrs, F-38044 Grenoble, France

**Keywords:** ClpXP, AAA+ ATPase, protease, unfoldase, structural biology, Cryo-EM, X-ray crystallography, NMR

## Abstract

AAA+ ATPases are ring-shaped hexameric protein complexes that operate as elaborate macromolecular motors, driving a variety of ATP-dependent cellular processes. AAA+ ATPases undergo large-scale conformational changes that lead to the conversion of chemical energy from ATP into mechanical work to perform a wide range of functions, such as unfolding and translocation of the protein substrate inside a proteolysis chamber of an AAA+-associated protease. Despite extensive biochemical studies on these macromolecular assemblies, the mechanism of substrate unfolding and degradation has long remained elusive. Indeed, until recently, structural characterization of AAA+ protease complexes remained hampered by the size and complexity of the machinery, harboring multiple protein subunits acting together to process proteins to be degraded. Additionally, the major structural rearrangements involved in the mechanism of this complex represent a crucial challenge for structural biology. Here, we report the main advances in deciphering molecular details of the proteolytic reaction performed by AAA+ proteases, based on the remarkable progress in structural biology techniques. Particular emphasis is placed on the latest findings from high-resolution structural analysis of the ClpXP proteolytic complex, using crystallographic and cryo-EM investigations. In addition, this review presents some additional dynamic information obtained using solution-state NMR. This information provides molecular details that help to explain the protein degradation process by such molecular machines.

## 1. Introduction

Macromolecular machines are dynamic assemblies composed of multiple protein components that form a structural scaffold, undergoing coordinated inter-subunit conformational remodeling in order to perform a wide range of crucial cellular functions. Probing the extent of these conformational changes and the associated dynamics is a prerequisite for deciphering the mechanisms that control how these macromolecular machines fulfill their functions at the molecular level and represents a major challenge for conventional structural characterization methods. Indeed, intrinsic features of these macromolecular complexes—such as their size, complexity, structural heterogeneity, and dynamics—generally hamper access to a complete global picture of the continuous process underlying their function. For instance, methods such as cryo-EM and X-ray crystallography capture data from samples trapped in static states, either frozen on grids or packed into crystal lattices, providing isolated snapshots of lower-energy intermediate conformations [1,2]. However structural characterization of several transient states is required to propose a model for the mechanism occurring in these complexes. While these snapshots are extremely valuable to obtain initial information about multi-protein complexes, the proper description of the continuous event occurring in the machinery requires the ability to probe the whole range of motions in these macromolecular motors over time. NMR is a widely used method to study dynamics at different time scale [3]. However, this technique long remained restricted to small domains or monomeric proteins. The recent outstanding progress in the experimental technics and development of specific sample labeling expanded the scope of this application to more complex targets of higher molecular weight, bringing access to new dynamic information on such machineries [4,5,6,7]. These new data could help to bridge the gap between the different snapshots and complete the structural data obtained from cryo-EM and X-ray crystallography [8].

The ATPases associated with diverse cellular activities (AAA+ ATPases) constitute a family of high-molecular-weight proteolytic machines, using a dynamic and complex mechanism involving major conformational changes to support their function [9]. AAA+ ATPases still represent a serious challenge for in-depth structural studies. Here, we focus on ClpXP protease to illustrate how an integrative approach can be adopted to study such intricate macromolecular machinery and ultimately leverage technical issues intrinsic to each independent method. The ClpXP degradation machinery encloses ClpX AAA+ ATPase acting as an unfoldase and the protease ClpP [10]. Early crystallization attempts on both isolated component of ClpXP, either under monomeric or oligomeric state, allowed researchers to experimentally solve the crystal structures of ClpP and then ClpX from different organisms (Figure 1) [11,12,13,14,15,16,17]. The resolution of these structures was a decisive factor in the structural characterization of the AAA+ protease family, as it enabled the visualization of protein folding along with a clear depiction of subunit interactions within the ring. Nevertheless, structural artifacts likely resulting from crystal contacts, as well as the absence of crystal structures of the full ClpXP complex, have largely limited our understanding of the function and regulation of the ClpXP AAA+ protease. Additionally, crystallization attempts on the ClpXP complex in the presence of a substrate protein also remained unsuccessful. Consequently, key mechanistic events—such as adenosine triphosphate (ATP) binding and hydrolysis, which fuel the mechanical force and concerted conformational changes across subunits of the complex driving substrate unfolding, translocation, and degradation—have remained out of reach to structural biologists.

Despite the availability of a low-resolution cryo-EM structure in the early 2010s [31], no significant progress was made on the ClpXP complex for nearly a decade, until recent advances in cryo-EM (Figure 1). These advances enabled the determination of near-atomic resolution structures of AAA+ proteolytic assemblies trapped at different stages of substrate processing, highlighting a conserved core mechanism [18,19,32,33,34]. The available structures of substate-bound AAA+ proteases constitute a unique opportunity to get access to molecular subunit–substrate interactions. This review aims to gather the breakthroughs brought by structural studies of ClpXP as a model system for the AAA+ protease superfamily to illustrate how the different structural biology techniques can be used synergistically to provide a better understanding of such high-molecular-weight dynamic macromolecular machines. Recent advances that have overcome the remaining bottlenecks preventing the deciphering of ATP binding and hydrolysis events will be presented. These steps are responsible for large-scale conformational changes driving substrate unfolding and proteolysis.

## 2. ClpXP, a Prototype for the AAA+ Proteolytic Machine Family

The family of AAA+ ATPase comprises molecular machines performing a wide range of essential cellular processes including proteolysis, protein remodeling, DNA replication, ribosome assembly, and viral replication [9,35,36,37,38,39,40]. AAA+ proteins share a common AAA+ ATPase domain consisting of a hexameric arrangement of monomers in a pseudo-helicoidal ring, forming a central axial channel, and are known to convert chemical energy from the hydrolysis of ATP into mechanical energy through conformational changes (Figure 2). Among these molecular machines, the AAA+ proteases can assemble into one or two AAA+ ATPase rings stacked on each side of a self-compartmented peptidase [9,20,38,41,42,43]. Upon binding of a substrate protein, the AAA+ ATPase ring exploits the ATP hydrolysis and the subsequent release of inorganic phosphate (Pi) and adenosine diphosphate (ADP) to drive the mechanical conformational changes required to unfold the substrate into an extended polypeptide chain and promote its translocation to the proteolysis chamber. This tight inter-subunit cooperation ensures a reliable translocation mechanism, restricting access to the protease only for recognized and properly unfolded proteins (Figure 2) [42].

ClpXP is a well-characterized macromolecular machine belonging to the AAA+ superfamily, dedicated to the unfolding and proteolysis of degradation-tagged protein substrates [44,45]. ClpXP activity participates in maintaining cellular homeostasis and quality control in eubacteria and mitochondria of eukaryotic cells [46,47,48]. This machinery proceeds by targeting incomplete proteins resulting from abortive translation, as well as damaged proteins [38,40]. Beyond its well-known role in protein quality control, recent advances uncovered that ClpXP is also involved in several highly regulated cellular processes, including bacteria cell division [49,50], mitochondrial gene expression [51,52], and respiratory chain function maintenance [53,54]. Over the three last decades, ClpXP has emerged as a prototype for other ATP-dependent proteases—such as ClpAP, ClpCP, HslUV, Lon, FtsH, PAN/20S, and the 26S proteasome—to investigate the operating principles governing substrate processing. The proteolysis machine consists of two coaxially stacked homoheptameric ClpP rings topped with one or two homohexameric ClpX complexes to form single- or double-capped ClpXP proteolytic assemblies, respectively, both competent for the unfolding and degradation of well-folded client proteins [18,55].

To achieve specific protein degradation, the ATP-bound ClpX hexamer recognizes client proteins with unstructured N-ter or C-ter or tagged with a specific degradation motif called a “degron” [19,38,56] and uses cycles of ATP hydrolysis to unfold and translocate these substrates through the central pore of the complex into the proteolytic ClpP barrel (Figure 2) [10,33,55]. Remarkably, the activity of ClpP alone is solely restricted to the proteolysis of small peptides diffusing in the degradation chamber, thus preventing uncontrolled proteolysis of larger proteins by free ClpP tetradecamer [57,58].

## 3. Structural Studies of Tetradecameric ClpP Protease

ClpP protease has been discovered as a heat shock protein [59,60]. Although this protein is constitutively present in the cell, it is overexpressed under stress conditions to allow the degradation of damaged proteins [46]. Crystallographic studies held in the late 1990s and the early 2000s provided the first fundamental structural information necessary for understanding the function and regulation of ClpP protease [12]. These studies highlighted a well-conserved structural organization among the various ClpP orthologs, characterized by a three-domain fold enclosing a dynamic N-ter region, a globular head, and a long α-helix handle domain [14,61]. Due to its enhanced stability, one structure of the ClpP heptamer ring has been solved rapidly, constituting the first evidence of a supramolecular organization of the protease (Figure 3).

### 3.1. Structural Insights into ClpP Catalytic Site

In all species, ClpP protomers typically assemble into a self-compartmentalized tetradecameric protease, constituted by two heptameric ClpP rings stacked together through interactions mediated by the handle region (Figure 3a). The core chamber encloses 14 canonical catalytic domains, conserved among the orthologs, each based on a Ser-97, His-122, and Asp-171 catalytic triad localized at the interface of two ClpP domains to perform the nucleophilic attack of the serine side chain [12,62]. The co-crystallization of ClpP heptamers with Z-Leu-Tyr-ChloroMethylKetone (Z-LY-CMK, Figure 4), and other commonly used serine protease inhibitors able to covalently bind the catalytic site, allowed, for the first time, for identifying and mapping the detailed interactions existing between the substrate side chains and the extended active site of the ClpP barrel [17,63,64,65,66]. The structural characterization discloses a complex adopting a tetrahedral conformation that mimics an intermediate state of the peptide cleavage, in which S97 and H122 are engaged in covalent bonds with the substrate, while the third catalytic residue, Asp171 is contacting the side-chain nitrogens of His122 and His138 from the adjacent monomer through hydrogen bonds. Completing the tetrahedral complex, the hemiketal oxygen atom of the inhibitor is positioned in the oxyanion hole, which is formed by the backbone amide nitrogens of Gly68 and Met98 (Figure 4) [17].

### 3.2. Structural Insights into the Gating Mechanism of the ClpP Axial Pore

In many crystallographic structures, the N-terminal domain of ClpP is unstructured with missing electron density at the extremities, reflecting the critical intrinsic flexibility of the region. However, some crystal structures have suggested that N-ter loops could be observed either under an “up” or a “down” conformation [14,16,17,67]. In the up conformation, the N-ter region adopts a β-hairpin loop, in which the seven first residues form the axial pore, while residues 8 to 16 adopts a flexible loop extending out of the apical surface of ClpP, controlling the opening of the axial pore (Figure 5a) [68]. Conversely, in the “down” conformation, no protruding loop is observed from the apical surface, and N-ter regions are poorly defined and oriented perpendicular to the seven-fold axis [14].

However, the conformational transitions of the N-terminal end observed in the ClpP protomer structure result from differences in the crystal packing environment of each individual ring (PDB code: 1YG6) [14]. More specifically, residues 8–11 in the apical loop in the “up” conformation are shown to share crystal contacts with residues from side surfaces of the adjacent ClpP tetradecamer. In contrast, the “down” N-ter conformation appears to be crystal contact-free and able to extend toward the lumen of the pore, thus clogging the axial pore. This observation raised the question of whether the “up” conformation might be an artifact resulting from crystal contacts [68,69]. The integration of solution NMR, cryo-EM, and molecular dynamics has established that in the wild-type (WT) apo ClpP, the N-terminal β-hairpin is well formed but exhibits conformational heterogeneity [70,71]. A network of hydrophobic contacts between the hairpin residues (Pro-5, Val-7, and Ile-20) and Leu-25 and Phe-60 of the adjacent protomer participates in holding the N-ter β-hairpins together to form the pore of *S. aureus* ClpP. Nevertheless, in absence of ClpP activation by ClpX, the pore diameter remains too narrow for the diffusion of large proteins, allowing only small peptides to be degraded. Individual mutations in any of these hydrophobic residues result in complete unfolding of the N-terminal β-hairpin structure and the loss of catalytic activity, even for small model peptide substrates [71].

Tremendous achievements have been reached in the modulation of AAA+ protease proteolytic compartment access through the identification of ClpP activators known as acyldepsipeptides (ADEPs). Co-crystallization trials with ADEP compounds have demonstrated that these molecules act on the protease compartment by binding the hydrophobic pockets recognized by ClpX to activate the protease (Figure 5) [21]. The binding of ADEPs mimics the interaction of active ClpP with ClpX [45]. ADEP molecules stabilize the N-ter regions in a rigidified β-hairpin conformation, thus allowing an opening of the axial pore to a diameter of ~20 Å. This conformational change enables the ClpX-independent entry of large substrates in the proteolysis chamber (Figure 5b). Stabilization of ClpP N-ter loops in the “up” position appears to be mediated through interactions between four charged residues. The Glu8 residue forms polar contacts with the side chain of Lys25 on the α-helix, locking the β-hairpin conformation at the N-ter of *E. coli* ClpP globular domain, thus removing the loop from the axial lumen. Additionally, the N-ter loops from two adjacent monomers are stapled together through intermolecular interactions between Glu14 and Arg15 to stiffen the edge for the axial pore (Figure 5c). The structural mechanisms underlying ClpP protease activation upon binding of ADEP analogues have been explored using methyl-TROSY NMR experiments, focusing on the relaxation properties of methyl groups (CH_3_) in *N. meningitidis* ClpP, specifically labeled to introduce isolated ^13^CH_3_ probes in either the axial loop or the handle region. ADEP binding induced changes in the TROSY spectra, suggesting that the axial loop undergoes a rigidification during activation [67], in agreement with the observations from crystallographic structures.

### 3.3. Conformational Switch of ClpP Handle Domain Controls the Catalytic Activity

While substrate protein entry is regulated by the axial pore, structural rearrangements of the N-terminus β-hairpins are not the sole regulator controlling ClpP activities. The proteolytic machinery has been crystallized in different conformations [72,73,74]. It is well established that the “extended” conformation, with an intact handle helix that ensures optimal positioning of the catalytic triads, represents the proteolytically active conformation (Figure 6). However, ClpP also crystallizes in inactive “compact” and “compressed” conformations. In both cases, the structure of the handle helix end and the handle strand are disrupted, altering the conformation of the catalytic triad. While the binding of activators such as ADEPs or benzoyl-leucyl-leucine (Bz-LL) stabilized the extended active conformation, it was unclear whether the compressed/compacted forms were sampled during the functional cycle [75]. Solution NMR has shown that at physiological pH, both the inactive and active forms of ClpP are sampled and in equilibrium in solution [76]. The extended form is stabilized by a salt bridge involving a histidine residue from the handle helix with an aspartic acid residue from the catalytic triad of an adjacent monomer. Protonation of the histidine has been proposed to induce unmasking of the handle helix end combined with the switch to the compact form. In addition, solution-state NMR studies of ClpP in the presence of a substoichiometric amount of activator established an allosteric shift between compact inactive and extended active conformations involving intra- and inter-ring communication [75]. Interestingly, the existence of a dynamic pore on the ClpP barrel side was suggested in an NMR study conducted in 2005 [77]. However, clear structural insights into these pores had yet to be determined. A few years later, structural studies on ClpP highlighted a compressed structure in which lateral pores appear large enough to accommodate the release of the degraded peptide [72,74].

## 4. ClpX Hexameric AAA+ ATPase, a Challenging Target for Structural Biology

### 4.1. First ClpX Crystal Structures

Due to its poor stability and high propensity to aggregate in solution, ClpX has long remained a challenge for structural characterization by X-ray crystallography. As a result, more than 10 years separate the resolution of ClpP tetradecamer [12] and the first crystal structure of the hexameric AAA+ ATPase ClpX ring [13]. In addition, mutagenesis and construct optimization were required to stabilize ClpX oligomeric structures. For example, the previously cited X-ray structure of the hexameric ClpX ring was obtained using an engineered single-chain ClpX hexamer to form and characterize the complex. Biochemical studies have shown that the ClpX N-terminal domain, which is connected to the large AAA+ domain via a highly flexible linker, binds the auxiliary tags of certain substrates and adaptor proteins but is not required for ClpXP degradation of ssrA-tagged substrates [80,81]. Indeed, the engineered AAA+ unfoldase pseudo-hexamers composed of genetically tied variants of *E. coli* ClpX lacking the N-terminal domain (ClpX^ΔN^), in combination with ClpP, form a complex able to degrade ssrA-tagged proteins with kinetic parameters comparable to those of wild-type ClpXP [81,82,83]. Consequently, most of the structural studies on the AAA+ unfoldase were carried out on ClpX^∆N^ variants to overcome the high flexibility of the N-ter region, which highly hinders crystallization of ClpX.

ClpX^ΔN^ structural data reveal that each unit folds into a large AAA+ domain (~260 residues) and a small AAA+ domain (~100 residues) (Figure 7), which, in the wild-type protein, are preceded by an N-terminal domain (~60 residues) [13]. The large and small domains together formed the AAA+ module, a structural feature common in many AAA+ enzymes, whereas the flexible N-terminal domain is only found in the ClpX family and remains as unresolved structure [22,80,81,84]. Concerning the supramolecular organization, these structures show that the large AAA+ domain of one subunit interacts with the small AAA+ domain of the neighboring subunit in a rigid-body manner [11,13,85].

The large AAA+ domain contains three loops, pore-1, pore-2, and RKH loops, respectively, named based on their sequence. These loops line the axial channel of the hexameric ring (Figure 7a,b), although the weak electron density suggests a large flexibility of these regions. Interestingly, the RKH loop is unique to the ClpX family, whereas pore-1 and pore-2 elements are common among AAA+ proteases and protein-remodeling machines. Additionally, the large AAA+ domain harbors an IGF loop pointing downward of the ClpX structure to mediate docking with ClpP (Figure 7b) [86].

### 4.2. Identification of the ClpX Nucleotide Binding Site

Resolution of the ClpX hexamer crystal structure allowed for the localization of the nucleotide binding site in the hinge between the large and small AAA+ domains of each protomer (Figure 8a). This site involves key residues, named box-II (I79), Walker-A (T126 and E130), Walker-B (D184 and E185) of the large domain, and sensor-II regions (R379) located in the small domain and the arginine finger (R307) from the large domain of an adjacent subunit (Figure 8b,c) [13]. Noteworthy, the trans-acting arginine fingers have been demonstrated to be able to sense the nucleotide state of the neighboring protomer, adopting distinct conformation depending on whether ATP or ADP is bound to the subunit. Such inter-subunit cooperation and signaling of hydrolysis between monomers have also recently been shown for the ClpX-like protein p97 [87].

Despite the important information provided by the crystal structure of the single-chain ring ClpX^ΔN^, many points remain unresolved. Firstly, in the available structures, two of the six subunits of this crystallized ClpX hexamer fail to bind nucleotides, resulting in aberrant rotation of the large AAA+ domain relative to the small domain, and the pore-1, pore-2, RKH, and IGF loops are generally disordered (PDB code: 3HWS, 4I81, and 4I4L) [13,85]. Furthermore, single-chain ClpX hexamers exhibit inappropriate spacing of IGF loops to properly dock AAA+ subunits in the multiple clefts at the surface of the ClpP heptamer [13,85]. Despite numerous attempts, none of the many AAA+ protein X-ray crystallography studies has successfully resolved the structure of a substrate protein engaged in these translocases. Consequently, the structural rearrangements required for ClpX function and the mechanisms underlying protein translocation remained obscure. These limitations illustrate the critical need for complementary methods to capture less stable and dynamic large complexes.

## 5. Cryo-EM Investigations Revealed the High-Resolution Structure of Active ClpXP Machinery

The structural insights into the interaction between ClpX and ClpP and the inherent open/closed conformations of the ClpP axial channel were initially derived mainly from crystal structures of ClpP [14] and a low-resolution 3D structure of ClpP bound to ClpA obtained by cryo-electron microscopy (cryo-EM) [23]. Despite the cryo-EM studies confirming that the access of the self-compartmented ClpP is regulated by the binding of the AAA+ unfoldase, the low resolution of the map prevented the identification of the key amino acids of the N-terminal region of ClpP involved in stabilizing the open conformation. Notably, it remained unclear whether this conformation was induced by docking of the IGF loops onto ClpP surface or through the axial interactions between the N-ter regions of ClpP and the axial loops of ClpA. The advances in detector technology and image processing have drastically improved the resolution that could be achieved by cryo-EM [1,88]. These developments often referred to the “resolution revolution” yielded high-resolution structures of very large, flexible biomolecular complexes that often resisted crystal structure determination for decades, providing impactful insights on the apprehension of the AAA+ superfamily. In a span of 5 years, numerous AAA+ machineries have been fully resolved at an atomic or near-atomic resolution [43] (Figure 1). Conversely to crystal structures, these AAA+ complexes structures obtained by cryo-EM are depicted in a substrate-engaged conformation, providing remarkable insights on the conserved mechanism by which ATP powers substrate translocation.

### 5.1. Insights into the ClpP/ClpX Interaction

High-resolution cryo-EM reconstruction of the ClpXP proteolytic machine [18,32,33,34,55] brought, for the first time, a description of the interaction mode of ClpX with the self-compartmented ClpP, confirming the docking model suggested by crystallographic structures (§ 4.a) (Figure 9a). The ring-shaped hexamer of ClpX anchors on a flat barrel of ClpP through the insertion of ClpX IGF loops inside hydrophobic grooves located at the external edge of the upper proteolytic machine. Specifically, the IGF motif (Ile268-Gly269-Phe270, Figure 9b) located at the tip of the loops inserts deeply into hydrophobic clefts of ClpP. In addition of these hydrophobic interactions, polar contacts are established between ClpX IGF loop residues Gly269, Phe270, Ala272, and the ClpP surface (Figure 9c). While these interactions are identical across all ClpX protomers, the Glu263-Gly267/Thr273-Ala276 residues, which are not directly contacting ClpP, adopt a wide structural diversity. This reflects the significant flexibility of the IGF loops, facilitating the adaptation of the asymmetric spiral of ClpX onto the flat surface of ClpP (Figure 9d,e). This mode of interaction suggests the existence of a modulation in ClpX/ClpP binding during the full degradation process.

In agreement with the observations from ClpP structures co-crystallized with ADEP compounds (§ 3.b), the interactions of IGF loops with ClpP highlighted by cryo-EM studies demonstrate that ClpX anchoring participates in the stabilization of ClpP N-terminal region in a “up” β-hairpin conformation. This conformation corresponds to a 20 to 30 Å pore diameter at the entry of the proteolytic compartment, subsequently allowing the substrate’s translocation toward the proteolytic chamber (Figure 9d) [21,33]. Noteworthy, such a pore size would be large enough to enable both the substrate translocation to the degradation machine and the release of proteolysis products. Most electron density maps confirm the docking of all the IGF loops of the ATPase hexamer within six of the seven hydrophobic pockets of ClpP, leaving one empty cleft at the surface of the chamber (Figure 9d) [19,33,55]. On the other hand, in one of the cryo-EM maps, the electronic density is missing for two consecutive IGF loops that could suggest a modulation of the IGF loop flexibility among the ClpX ring [55]. The authors proposed a model in which the flexibility of the IGF loops could help to adopt an “extended/stretched” conformation, which likely causes strain and facilitates pore loop release and subsequent binding to the adjacent site on the translocated protein. Based on those results, a model in which the ClpX hexameric spiral could rotate with respect to ClpP in response to an ATP hydrolysis event was proposed [55]. However, independent experimental data are missing to support this model.

### 5.2. ClpX Structural Rearrangements upon Substrate Binding

The cryo-EM structures of *E. coli*, *N. meningitidis*, or *L. monocytogenes* ClpX^ΔN^ bound to ClpP solved at near-atomic resolution allowed for clearing some ambiguities that were remaining regarding the process of protein degradation [18,19,33,34,55]. Compared to existing crystallographic structures previously discussed, cryo-EM allowed access to the structure of the ClpXP complex with a protein substrate engaged at the entry of the axial pore. In the ClpX substrate-free configuration, the ClpX rings remain as flat hexamers (Figure 10a), as described in previous ClpX/ClpP structural studies [13,18]. Substrate binding seems to be determinant for the formation of the downward spiral-shaped conformation adopted by the ClpX ATPase ring, which enables substrate translocation into ClpP proteolysis compartment (Figure 10a,b,f), thereby confirming the relationship between substrate processing and the helical organization previously proposed [18]. Cryo-EM structures obtained on all types of substrate-engaged AAA+ ATPases revealed that the spiral-shaped conformation wrapped around the substrate is shared among the AAA+ family [89,90,91]. The presence of the substrate engaged in the central pore of the ClpX ring as well as ClpP on the bottom side of the ClpX ring, respectively, stabilize the RKH, pore-1, and pore-2 axial loops and the IGF loop, allowing these motifs to be fully defined in the structures (Figure 10c). As expected, the central channel occupied by the translocated substrate is lined solely by the RKH, pore-1, and pore-2 loops of the large subdomain of the ATPase (Figure 10d). Confirming the X-ray structures, the nucleotide-binding pockets are found at the interface between adjacent subunits, involving both the surfaces of a small domain of a given subunit packed with the large subdomain of the clockwise neighboring protomer, resulting in a rigid-body subunit. Consequently, the functional ring can be viewed as six rigid-body subunits, composed of neighboring large and small AAA+ domains connected by short hinges that link the two domains of each AAA+ module (Figure 10e).

### 5.3. ClpX Substrate Recognition

To prevent abortive protein translation by the ribosome, all cells employ quality control mechanisms to ensure misprocessed proteins degradation and to maintain proteome integrity. In *E. coli* and other eubacteria, the tmRNA system adds an ssrA degron to the C-terminus of the aberrant proteins, subsequently resulting in their recognition and degradation by the AAA+ ClpXP protease [56,92]. Proteins bearing this 11-amino-peptide ssrA degron (AANDENYALAA-COO^-^ in *E. coli*) [93] are recognized and degraded by numerous proteases such as ClpXP and ClpAP [94,95]. Studies on the protein recognition mechanism have identified that the two terminal residues -Ala-Ala-COO^-^ of the ssrA tag are essential for the early interaction with ClpX. The high-resolution map of ClpXP allowed for solving a so-called “recognition complex” capturing the initial steps of substrate insertion into ClpX (Figure 11) [19]. The C-ter part of the degron is taken over by residues Arg228, His230, and P231 from the RKH loops of protomers #B, #C, and #D, as well as by residues Tyr153 and Val154 from the pore-1 loops of ClpX subunit #A and #B, while the A-A-COO^-^ is contacted by Thr199 and Val202 from the ClpX pore-2 loop from subunit #A.

### 5.4. Intermediate Complex Sheds Light on Substrate Translocation

One of the major advances provided by these high-resolution cryo-EM structures is the capture of the so-called “intermediate complex” in the event of substrate processing, thus providing new insights at a near-atomic resolution of the mechanism involved in the substrate gripping and translocation [19,34,55]. Indeed, these structures are depicting the highly conserved axial loops of ClpX adopting a helical staircase conformation taking over the substrate along the axial pore of the AAA+ ATPase. The mapped interactions between the loops and the unfolded substrate peptide are solely mediated by the insertion of the side chains of the conserved pore-1 loop Tyr153 and Val154 residues between β-carbons of every two residues on the opposite side of the extended peptide, similar to the teeth of a gear mechanism (Figure 12a). These steric interactions ensure ClpX has a sufficient substrate grip to apply the force required for the unfolding and translocation. Interestingly, contacts between the substrate and pore-1 loops can be established either by Tyr153 or the neighboring Val154, suggesting that the hydrophobicity of the loop is responsible for the interaction rather than the aromatic residue itself. The helical staircase along the substrate has also been observed in the high-resolution structures of substrate-bound AAA+ ATPases such as 26S proteasome (PDB code: 6MSD), Lon (PDB code: 6U5Z), Vps4 (PDB code: 6BMF), YME1 (PDB code: 6AZ0), and ClpB (PDB code: 6RN2) (Figure 12b–d), with potentially a Phe or a Trp amino acid instead of the pore-1 loop Tyr153 and a Lys amino acid instead of Val154 [89,90,91,96,97]. This demonstrates the conservation of a translocation mechanism among the AAA+ ATPase superfamily, which allows for the integration of a substrate regardless of the peptide sequence or the N-ter/C-ter orientation during the insertion. However, the interleaving mechanism of the substrate–pore loop 1 interactions also provide a trimming knob for the substrate to modulate these contacts within the central pore. Thus, bulky, hydrophobic, or aromatic residues of the substrate fit successfully between the pore loop aromatic residues, resulting in a strengthened grip on the substrate, while smaller residues of the substrate, such as glycine, which cannot fit between the loop residues, would decrease the gripping ability of the unfoldase.

The structures of the substrate-engaged complex (Figure 11 and Figure 12) could correspond to snapshots of the ssrA-mediated translocation at two consecutive time frames of the process [19]. Indeed, the ssrA degron initially engaged at the top of the axial pore in the recognition complex structure (Figure 11) progressed by 25 Å or six residues deeper into the channel in the intermediate complex (Figure 13a). Numerous experimental data show that this progression is concomitant with a structural reorganization of the pore-1 loops of different ClpX protomers to support the substrate translocation. The height of Tyr153 in pore-1 with respect to the plane defined by the tip of the IGF loops can be used as a reporter of structural rearrangement (Figure 13a). These representations allow for depicting, in the recognition complex, a downward spiraling staircase constituted by subunits #A to #E, while the position of subunit #F suggests that its pore-1 loop has risen to the level of subunit #D at an intermediate height (Figure 13b,c). By contrast, subunit #A of the intermediate complex appears to be lower with reference to the recognition complex one, while a similar downward spiraling staircase is now composed of subunits #B to #F (Figure 13d).

## 6. Models for ClpXP Substrate Translocation Mechanism

### 6.1. Coupling ATP Binding to Pore-1 Loop/Substrate Engagement

Previously resolved ClpXP structures of the recognition and intermediate complexes suggested the coexistence of different nucleotide states (Figure 13). In both of these ClpXP/substrate complexes, five of the six nucleotide binding sites are occupied by ATP, forming the downward spiral around the substrate. The sixth ClpX subunit is ADP-bound or nucleotide-free. These structures revealed that substrate contacts mediated by pore-1 loops are restricted to ATP-bound spiral subunits, while ADP-bound or nucleotide-free subunits are characterized by a pore-1 loop off centered from the axial pore and disengaged from the substrate. High-resolution structural analysis of ATP-bound and apo AAA+ ATPase subunits demonstrates that ATP binding induces a 15–40° rotation of the intra-subunit hinge between small and large AAA subdomains (Figure 14a). This rotation has been proposed to engage the pore-1 loop of the corresponding ATPase subunit on the substrate (Figure 14b) [13,19]. These structural features of ATP-loaded subunits were found in the different ClpXP structures resolved in the presence of substrate proteins and ATP [32,33,55] and seem to be conserved in different homologous AAA+ ATPases such as Lon, VAT, or 26S particle [24,25,91,98,99], thus highlighting a correlation between nucleotide site occupancy and the conformation of the pore-1 loop of the respective protomer.

### 6.2. A Conserved Mechanism for Substrate Pulling by the AAA+ Subunit

Numerous cryo-EM structures of substrate-bound AAA+ unfoldases suggest a conserved substrate translocation mechanism at the level of each AAA+ ATPase subunit [9]. It is proposed that each ATP hydrolysis event occurs within the lowest subunit. The loss of γ-phosphate inherent to ATP hydrolysis in the lowest subunit of the spiral destabilizes the ATP-dependent bonds made by arginine fingers between two adjacent subunits (Figure 8). Subsequently, ADP release is coupled to disengagement of pore-1 loop from the substrate (Figure 14b). Following this disengagement, pore-1 loop undergoes an important vertical rotation. This large conformational shift induces local rearrangements of the other ATP-bound subunits propelling the substrate toward the ClpP proteolytic cavity. Subsequent ATP binding re-engages the pore-1 loop with the substrate at the top of its pre-loop staircase. Each ATP hydrolysis cycle sets in motion a coordinated choreography among the different subunits of AAA+ ATPase and powers a hand-over-hand pulling of unfolded polypeptide through the central channel of the hexamer.

In the proposed mechanism, the hydrolysis of the ATP only occurs in the most basal subunit before ascending to the top of the helix. The helical structure of ClpX as a whole creates a ratchet mechanism. The displacement of the subunit is irreversible, preventing the system from moving backwards. Additionally, peptide hydrolysis and diffusion out of ClpP render substrate protein transfer from ClpX to ClpP irreversible.

### 6.3. Allosteric Regulation of Translocation Models

The functional cycle fueled by the hydrolysis of ATP is well established at the level of each individual subunit, but several models have been proposed to describe how the six subunits of ClpX cooperate together to ensure efficient translocation of the substrate. A widely accepted sequential model has been proposed for AAA+ unfoldase such as Yme1 [96], 26S-proteasome [91], VAT [99], and Lon [98], wherein a sequential ATP hydrolysis cycle powers a hand-over-hand pulling of unfolded polypeptide through the central channel of the hexamer [25,96,99,100,101]. In this model, the large vertical rotation of the pore-1 loop resulting from ATP hydrolysis and ADP release induces smaller rotations (5–10°) of ATP-bound subunits. Together, these collective rotations of ATP-bound subunits create a power stroke, allowing for substrate translocation progression of two amino acids towards ClpP protease [42,102]. The sequential hydrolysis of ATP at the extremity of the helical staircase results in unidirectional substrate translocation and the movement of pore loops along the substrate together with the counterclockwise hydrolysis cycle, subsequently resulting in a repeated downward pulling force on the substrate (Figure 15a,b). During a complete cycle of ATP hydrolysis, the subunit of ClpX, at the top of the hexameric ring, first engages its pore-1 loop in an interaction with the substrate to escort it into the axial pore. Then, the subunit of ClpX progresses through each position of the spiral while maintaining a close interaction with the substrate until reaching the lowest position of the spiral, where it detaches from the substrate after ATP hydrolysis and ADP release events.

While the sequential translocation model relies on a two-substrate-residue step translocation per ATP hydrolysis event, the structural snapshots captured during substrate processing, i.e., the so-called recognition (§5.c) and intermediate (§5.d) ClpXP structures, depict a substrate progression of 25 Å between the two steps. This observation suggests that a single ATP hydrolysis event leads to a six-residue-step translocation in that particular case. Since these two structures account for most tomograms, it is thought that these two conformations correspond to consecutive frames of the translocation process, and additional transient states of translocation with smaller steps are thus not likely to exist in solution. In the recognition complex, the pore-1 loops of all subunits are in contact with a clear helical staircase formed by the Tyr153 from #A to #E, except for the subunit #F, which is disengaged from the substrate where the Tyr153 from #F appears to be at mid-height of the staircase. Conversely, in the intermediate complex, the upper subunit #A, which was previously packed into the helical staircase, is now unpacked as Tyr153 descends. Simultaneously, the other subunits adopt the classical helical staircase conformation. The existence of unconventional translocation events seems to be reinforced by numerous optical trapping experiments showing that an elementary translocation step powered by ClpXP represents a 5- to 8-residue progression (Figure 15c) [103,104,105,106]. In order to explain such translocation steps in the axial pore of ClpXP, a probabilistic model of ATP hydrolysis has been proposed, in which a power stroke would be accompanied by a downward move of the upper subunit bound to the substrate to pull it six residues below. In this model, any ATP-bound subunit would be able to trigger ATP hydrolysis to position itself at the bottom of the hexameric spiral, causing a translocation of a distance depending on the initial position of the subunit in the complex [107].

## 7. Conclusions

Protein degradation systems play an important role in cellular homeostasis, regulating protein expression and function, and playing a key role in protein maintenance. These systems play decisive roles in diverse cellular processes, such as bacterial division and virulence, making them attractive targets to fight against cancer or pathogenic bacteria. Determining the 3D structures and complex mechanisms of these machines at the atomic resolution remains a major challenge in structural biology today. Recent advances in the cryo-EM method have significantly increased our ability to capture high-resolution structures of these machines in the heat of action. This review focuses on the bacterial and mitochondrial protein degradation systems, ClpXP. This machinery consists of two main components: the ClpP protease and the ClpX ATP-dependent unfoldase, a member of the AAA+ ATPase family. The structure of ClpP has been resolved by X-ray crystallography, and its mechanism is relatively well characterized. ClpP assembles into a barrel-shaped tetradecamer (14 subunits), and peptide cleavage occurs in the barrel through one of the 14 catalytic triads protected inside the barrel cavity. Several high-resolution structures and additional NMR data revealed that the loops at the N-terminus of ClpP subunits act as a gate to close the entry pore. The binding of ClpX hydrophobic loops or certain activator mimetic drugs to hydrophobic pockets on the ClpP surface induces conformational changes that can stabilize the N-ter loop in an open gate conformation, allowing for substrate entry, an essential step for processing large amino acid chains, such as unfolded polypeptides.

In contrast to ClpP, structural studies of ClpX are more challenging due to its inherent instability and flexibility, which make its purification and crystallization difficult. However, over the past five years, several high-resolution cryo-EM structures of ClpX stabilized by ClpP have been resolved. These ClpXP complex structures reveal an arrangement of ClpX in a hexameric ring weakly bound to ClpP 14-mer by hydrophobic loops. A key feature of the ClpXP assembly is the symmetry mismatch between the seven-subunit ClpP and six-subunit ClpX, resulting in one unoccupied ClpP hydrophobic binding pocket that may enable potential structural rearrangement between the weakly bound ClpX/ClpP rings. High-resolution cryo-EM structures of the ClpXP complex-processing substrates highlight some of the key steps involved in protein degradation. These snapshots highlight the major role of the two ClpX pore loops in substrate recognition and engagement within the channel, as well as the rearrangement of the ClpX ATPase ring into a helix that grabs the substrate and translocates it inside the ClpP catalytic barrel. While the ATP hydrolysis-powered mechanism of substrate pulling by each ClpX subunit is well understood, several models have been proposed to describe the cooperative mechanism between each of the six ATPase subunits enabling the translocation of unfolded chains into the ClpP proteolytic compartment.

Although cryo-EM provides high-resolution structures of the ClpXP machinery engaged in different stages of substrate processing, this approach only provides a static picture of the system and rarely reports the kinetic data needed for a complete understanding of the mode of action at atomic resolution. In the future, applications of advanced solution-state methyl-NMR approaches combined with biochemical studies are expected to provide complementary information to high-resolution cryo-EM structural data regarding the dynamic ballet and timing of each key step in the functional cycle of ClpXP machinery.

## Figures and Tables

**Figure 1 biomolecules-15-01097-f001:**
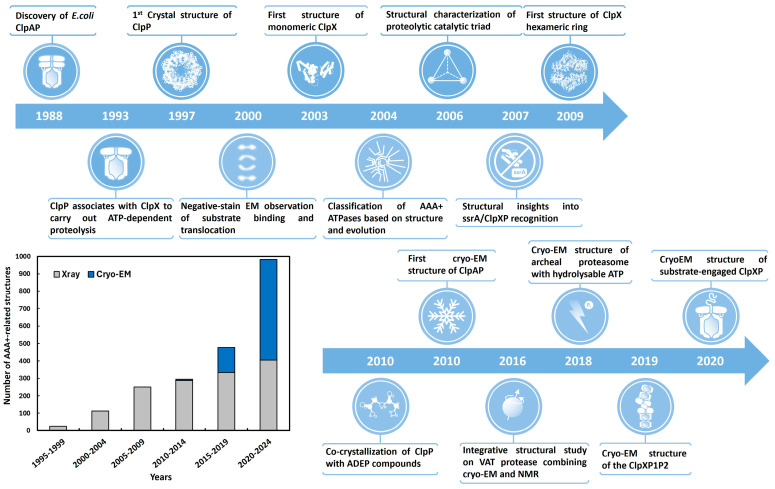
Timeline of the major biostructural advances leading to an improved mechanistic description of the substrate translocation and degradation by the bacterial AAA+ protease ClpXP [11,12,13,18,19,20,21,22,23,24,25,26,27,28,29,30]. Insert: number of AAA+ protease structures published over years using either X-ray (gray) and cryo-EM (blue) methods (from PDB data bank https://www.rcsb.org/ accessed on 1 January 2025).

**Figure 2 biomolecules-15-01097-f002:**
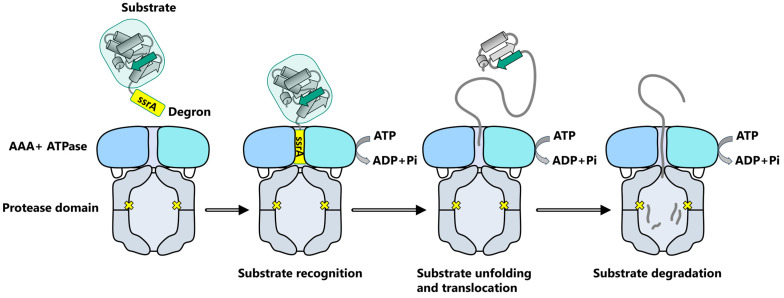
Summary of substrate degradation by ClpXP. The ssrA degron is located on C-Ter of the substrate protein. The substrate is represented in gray with a green background, the ssrA degron in yellow, and a selected region as the indicator in green. AAA+ ClpX unfoldase is represented with different shades of blue and ClpP protease in gray, with their catalytic sites represented as yellow crosses.

**Figure 3 biomolecules-15-01097-f003:**
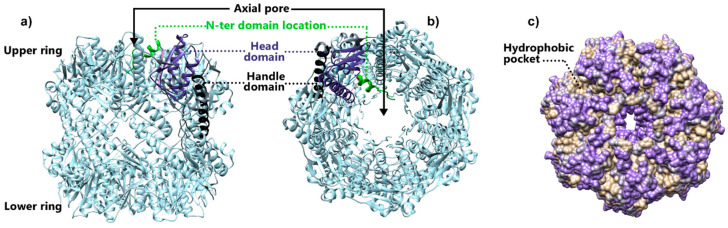
Structure of the ClpP tretradecamer of *E. coli* (PDB code: 1YG6) [14]. (**a**) Side view. A ClpP protomer has been highlighted from the complex with distinct colors corresponding to the three regions of ClpP. Gray: handle; purple: head; light green: location of the N-ter region. The lack of electronic density highlights the critical flexibility of the N-ter domain. The two upper-left protomers of the upper ring have been set transparent to reveal the different regions within a single protomer. (**b**) Top view of the ClpP tetradecamer. (**c**) Top view of the surface representation. The surface is color-coded according to hydrophobicity (purple: most hydrophilic; beige: most hydrophobic) to highlight the hydrophobic pockets involved in the interaction with ClpX.

**Figure 4 biomolecules-15-01097-f004:**
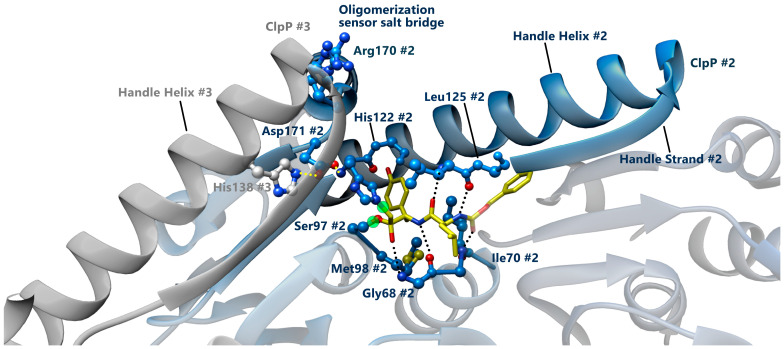
Structure of the Z-LY-CMK inhibitor bound to *E. coli* ClpP. Multiple bonding interactions between Z-LY-CMK (yellow) and the 3rd ClpP protomer of a heptamer are shown (PDB code: 2FZS) [17]. Hydrogen bonds between Z-LY-CMK and ClpP are shown as dashed lines. Gly68, Ile70, and Leu125 create four hydrogen bonds with the peptide backbone of the inhibitor, while the backbone amide nitrogens of Gly68 and Met98 form an “oxyanion hole” and make hydrogen bonds to the hemiketal oxyanion of the inhibitor. The inhibitor is attached to ClpP by two covalent bonds (green circle), one between the carbonyl carbon of the inhibitor and Oγ of Ser97 and the other between the methylene group of the inhibitor and Nε2 of His122. The so-called “oligomerization sensor” Arg170 form an inter-ring salt bridge that further help to correctly position the catalytic Asp171. The third catalytic residue, Asp171, forms a hydrogen bond with Nδ1 of His122, and an additional hydrogen bond is formed with Nε2 of His138 from the adjacent monomer (yellow dotted lines).

**Figure 5 biomolecules-15-01097-f005:**
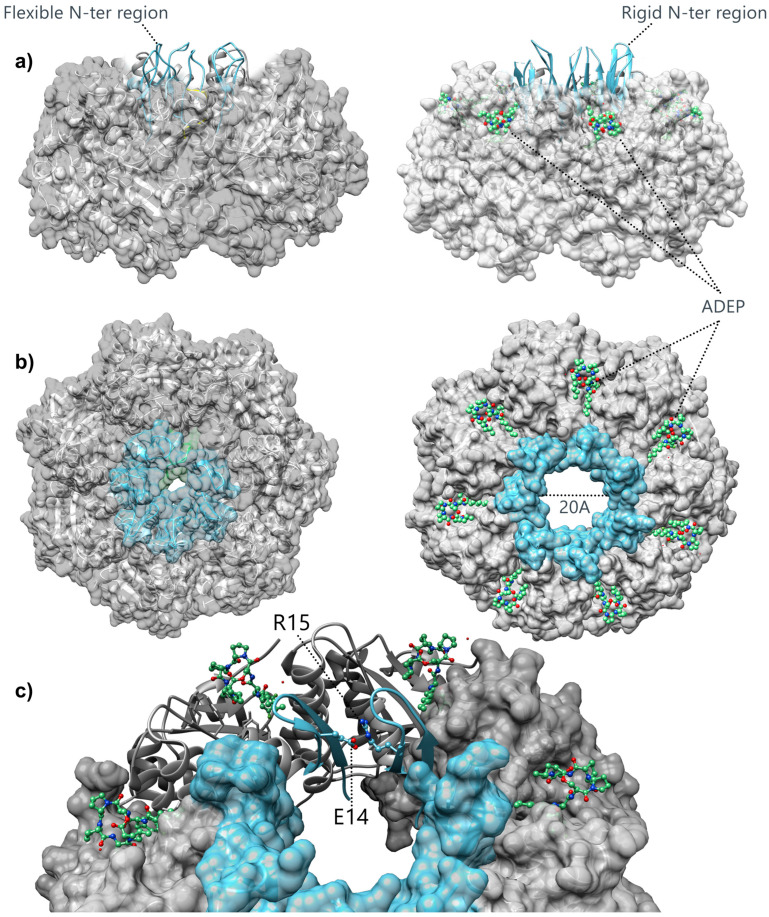
Crystal structure of *E. coli* ClpP in the apo form (PDB code: 1YG6) [14] and bound to acyldepsipeptide or ADEP (PDB code: 3MT6) [21]. (**a**) Side view of the crystal structures. Ribbon representation of the N-terminal region of ClpP protomer without ADEP (**left**) or bound to ADEP (**right**). The N-terminal region in the “up” conformation is shown in blue. (**b**) Top view of the crystal structure of ClpP (ADEP-free) (**left**) or bound to ADEP (**right**), leading to an opening of the pore of around 20 Å. (**c**) Close-up view of the residues 1–18 of two N-terminal regions shown as ribbons. The residues E14 and R15, which are involved in intermolecular interactions stabilizing the open conformation of the axial channel, are displayed in balls and sticks and labeled accordingly.

**Figure 6 biomolecules-15-01097-f006:**
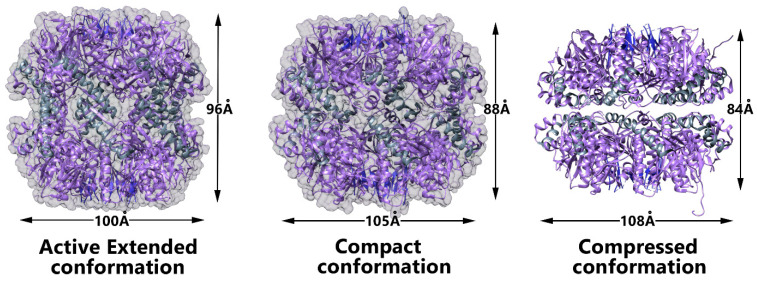
Crystal structures of *S. aureus* ClpP extended state (**left**) (PDB code: 3STA) [72,73,75,78,79], compact state (**middle**) (PDB code: 4EMM) [73], and compressed state (**right**) (PDB code: 3QWD) [72].

**Figure 7 biomolecules-15-01097-f007:**
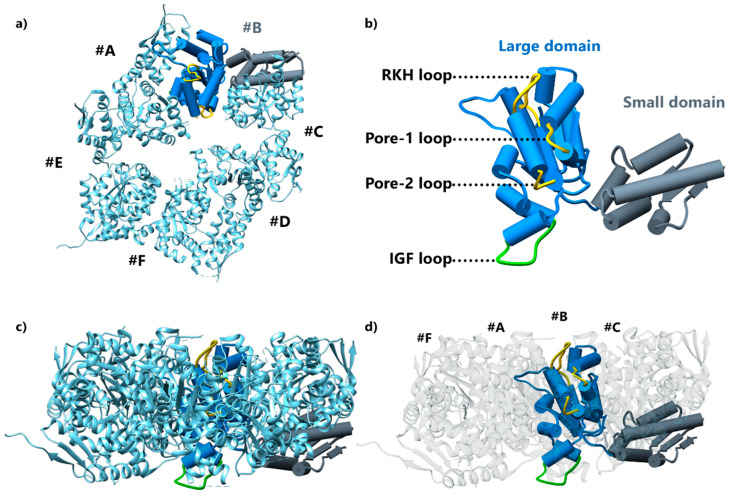
Structure of the monomer and hexamer of *E. coli* ClpX (PDB code: 3HTE). (**a**) Top view of a ClpX hexamer. The protomer #B is depicted in cartoon visualization with the large and small AAA+ domains, respectively, colored in blue and gray. (**b**) Cartoon visualization of a ClpX protomer. The poorly defined axial RHH, pore-1, and pore-2 are highlighted in yellow. The poorly defined IGF loop is highlighted in green. (**c**,**d**) Side view of a ClpX hexamer. In (**c**), the protomer #B is depicted as in panel (**b**) with the same orientation. The loops of the large domain are highlighted in the structure (IGF loop in green; RKH, pore-1, and pore-2 loops in yellow). In (**d**), the protomers #D and #E are set to 100% transparent, and protomers #A, #C, and #F are set to 50% transparent to allow for a proper view of the loops of protomer #B lining the axial pore.

**Figure 8 biomolecules-15-01097-f008:**
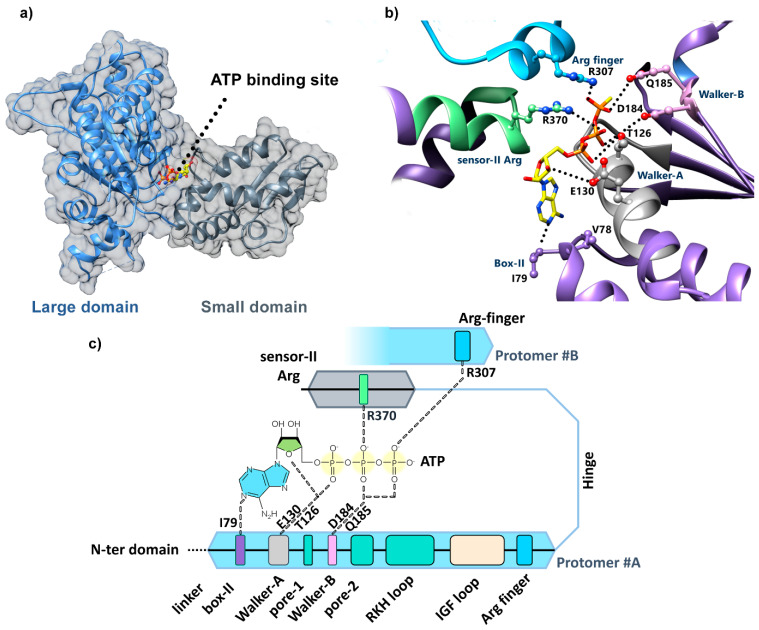
Crystal structure of ClpX^ΔN^ protomer bound to ATP. (**a**) The ATP binding pocket is located at the hinge between the large and small domains (PDB code: 3HWS). (**b**) Details of the ATP binding pocket in the ClpX hexamer (PDB code: 6WRF) with the conserved motifs and residues involved in the ATP binding including the sensor-II, BoxII, Walker-A, and Walker-B from a given protomer (#D in this example) and the arginine finger from the adjacent subunit (#E in this example). (**c**) Schematic of the ATP binding pocket involving key residues from the large (blue) and small (gray) domains of a given protomer, as well as residue R307 from the arginine finger from the adjacent subunit.

**Figure 9 biomolecules-15-01097-f009:**
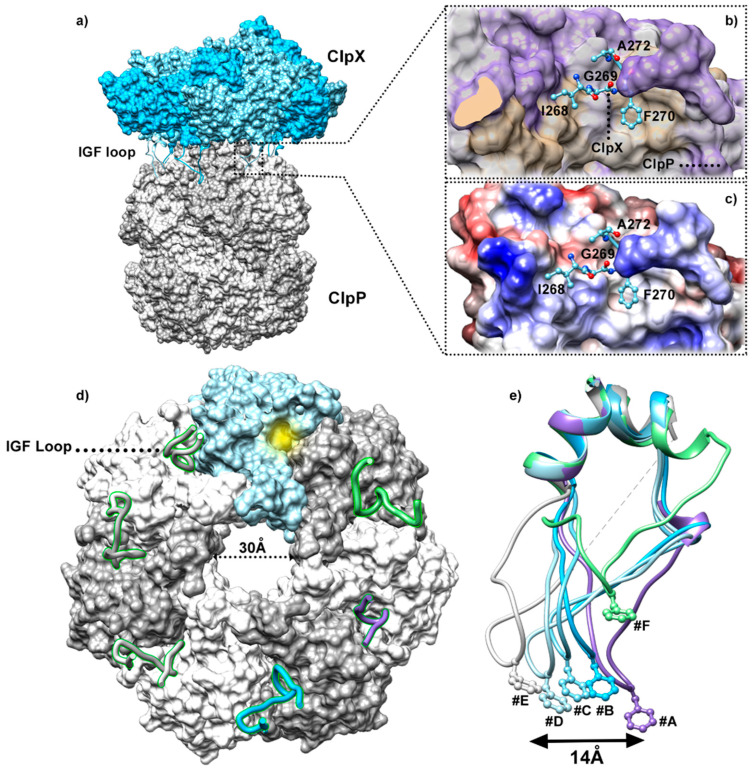
(**a**) Composite cryo-EM structure of ClpX bound to ClpP and protein substrate (PDB codes: 6PPE and 6PP6). The insets show a close-up view of the IGF sequence of ClpX bound deeply in a ClpP cleft. ClpP is represented as a surface and colored with respect to (**b**) hydrophobicity and (**c**) polarity. (**d**) Axial view of ClpP (PDB code: 6PPE) bound to ClpX showing an open axial pore, with the clefts that serve as docking sites for the ClpX IGF loops (displayed in ribbon). The ClpP protomer with an empty cleft missing an IGF loop is colored in blue, and the cleft is highlighted in yellow. (**e**) Structure alignment of the IGF tip loops, suggesting the flexibility adopted by the 6 loops to dock the ClpX asymmetric ring on top of the flat ClpP barrel.

**Figure 10 biomolecules-15-01097-f010:**
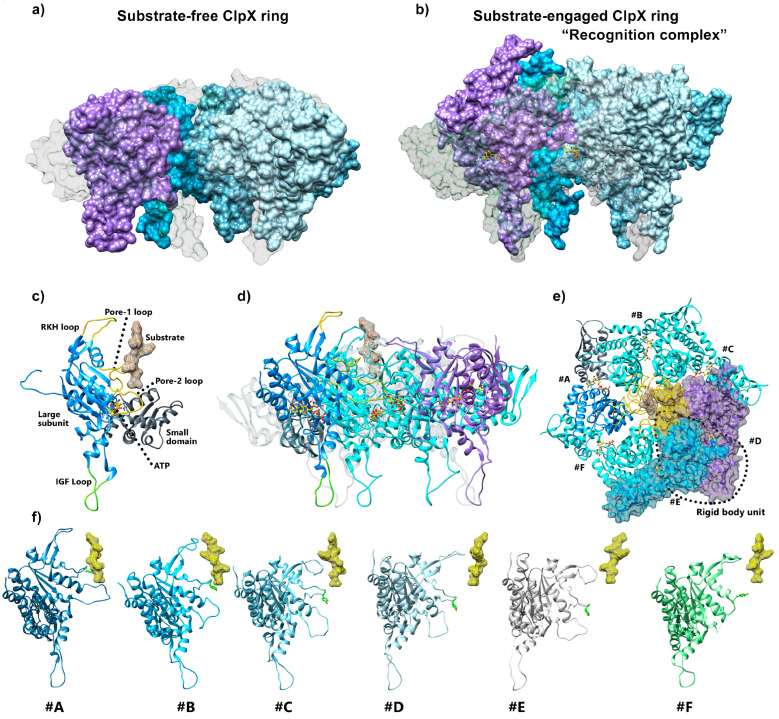
Cryo-EM structure of substrate-free and substrate-bound ClpX. (**a**) Flat substrate-free ClpX hexamer in surface representation colored by ClpX protomer (PDB code: 6SFW). (**b**) Substrate-engaged ClpX hexamer in surface representation colored by ClpX protomer (PDB code 6WRF). (**c**) Structure details of a ClpX protomer (#A) with substrate engaged. The large and small domains are colored in blue and gray, respectively. The axial loops (RKH, pore-1, and pore-2) are highlighted in yellow. The IGF loop is highlighted in green. The substrate is depicted in surface representation. (**d**) Side view of ClpX hexamer with substrate engaged. The colors are conserved for ClpX protomer (#A). Protomers (#D) and (#E) are set transparent to facilitate the observation of the inner part of the axial pore. (**e**) Top view of the ClpX hexamer. One of the rigid-body subunits involving the small domain of protomer (#D) and the large domain of protomer (#E) is circled with dotted line. (**f**) Representation of the six ClpX protomers interacting with the substrate, as observed in the cryo-EM structure of the substrate-engaged ClpX hexamer. The six protomers are represented with the same orientation as protomer (#A) in panel (**c**). The height of the substrate has been kept constant for the six protomers to highlight the differences in the positions of the axial loops from the six protomers.

**Figure 11 biomolecules-15-01097-f011:**
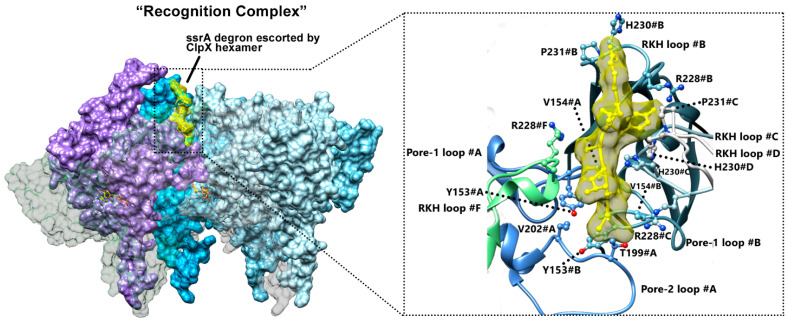
Recognition complex. Substrate-engaged ClpX hexamer in surface representation colored by ClpX protomer (PDB code: 6WRF). The two protomers from the front of the ring are set transparent to allow for a view of the recognized substrate. The substrate is displayed in yellow in the surface and atom representations. Insert: zoom on the molecular details of the ssrA degron recognition by ClpX involving the RKH and pore-1 loops of various protomers from the ring. The axial loops of the ClpX protomers are colored by protomers and displayed as ribbons.

**Figure 12 biomolecules-15-01097-f012:**
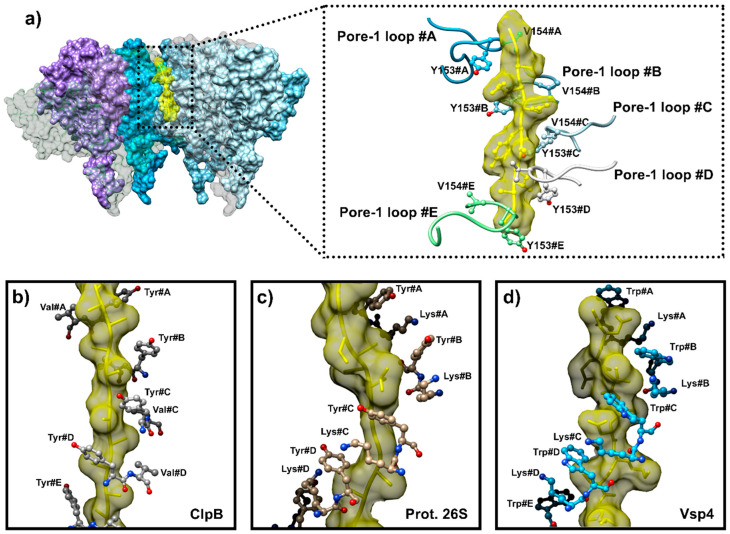
Helical staircase organization along the engaged substrate. (**a**) Intermediate complex: substrate-engaged ClpX hexamer in surface representation colored by ClpX protomer (PDB code: 6WSG). The two protomers from the front of the ring are set transparent to allow for a view of the recognized substrate. The substrate is displayed in yellow in the surface and atom representations. Insert: zoom on the molecular details of the interaction of engaged substrate with the ClpX protomers organized as a helical staircase around the substrate through the Tyr153 and Val154 residues of the pore-1 loop of each protomer. (**b**–**d**) Example of the helical staircase conformation conserved through the AAA+ ATPase family observed in the cryo-EM structure of substrate-engaged complex of (**a**) ClpB (PDB code: 6RN2), (**b**) 26S proteasome (PDB code: 6MSD), and (**c**) Vsp4 (PDB code: 6BMF). The amino acids inserted between the β-carbons of every two residues of the substrates are displayed in a ball and stick representation.

**Figure 13 biomolecules-15-01097-f013:**
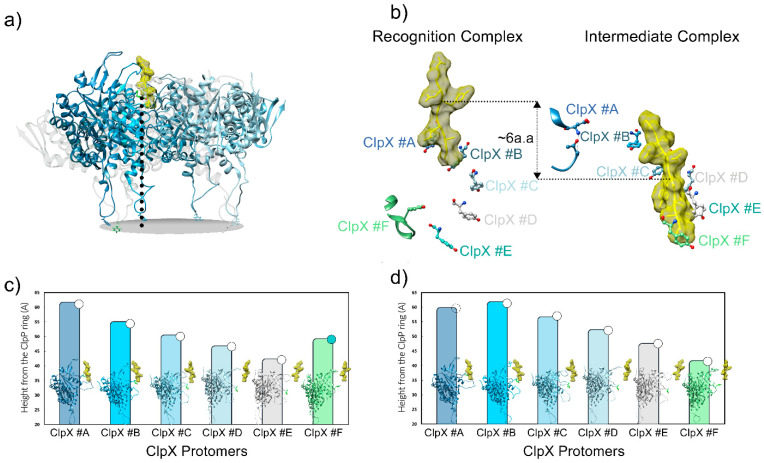
Structural variation of ClpX loops in recognition and intermediate complexes bound to substrates (PDB code: 6WRF and 6WSG). (**a**) The distance (dotted line) between the Tyr153 and the IGF loop/ClpP interface plan (gray plan) has been measured for each protomer in the recognition and intermediate complexes. (**b**) Six-amino-acid translocation step undergone by the substrate between the two ClpXP transient states. Tyr153 of each subunit is represented in balls and sticks, with Tyr153 of subunits #F and #A disengaged from the substrate, respectively, in the recognition and intermediate structures. (**c**) and (**d**) are representations of the height of Tyr153 with respect to the IGF/ClpP interface for all subunits in the recognition (**c**) and intermediate (**d**) complexes. The white, green, and dashed circles on top of the bars indicate if the ClpX protomer is ATP-free (dashed), ATP-loaded (white), or ADP-loaded (green).

**Figure 14 biomolecules-15-01097-f014:**
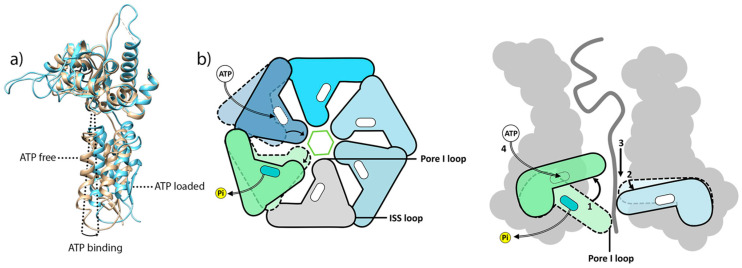
Substrate translocation driven by ATP hydrolysis. (**a**) Top view of superimposition of an ATP-free and ATP-loaded ClpX protomer from the intermediate complex ClpX ring after structural alignment of the large domains of the protomers (PDB code:6WSG). The dotted lines give the orientation of the small domain of the protomer in the ATP-free and ATP-loaded configurations. The curved arrow depicts the horizontal rotation of the small domain with respect to the large domain upon ATP binding. (**b**) Mechanism for processive substrate translocation driven by a complete cycle of ATP hydrolysis among the AAA+ ATPase family. Upon release of Pi, the green subunit, the substate-pore-1 loop interactions, as well as the inter-subunits interactions mediated by the Arg fingers and ISS loops are disrupted, leading to a disengagement of the subunit from the complex. Additionally, this release is combined with a 30° to 40° vertical rotation to reposition itself on top of the spiral, while the other subunits undergo a 5° to 10° rotation to follow the progression of the substate towards the protease chamber. The ATP-free subunit previously disengaged joins the helical staircase assembly upon the binding of a new ATP [91].

**Figure 15 biomolecules-15-01097-f015:**
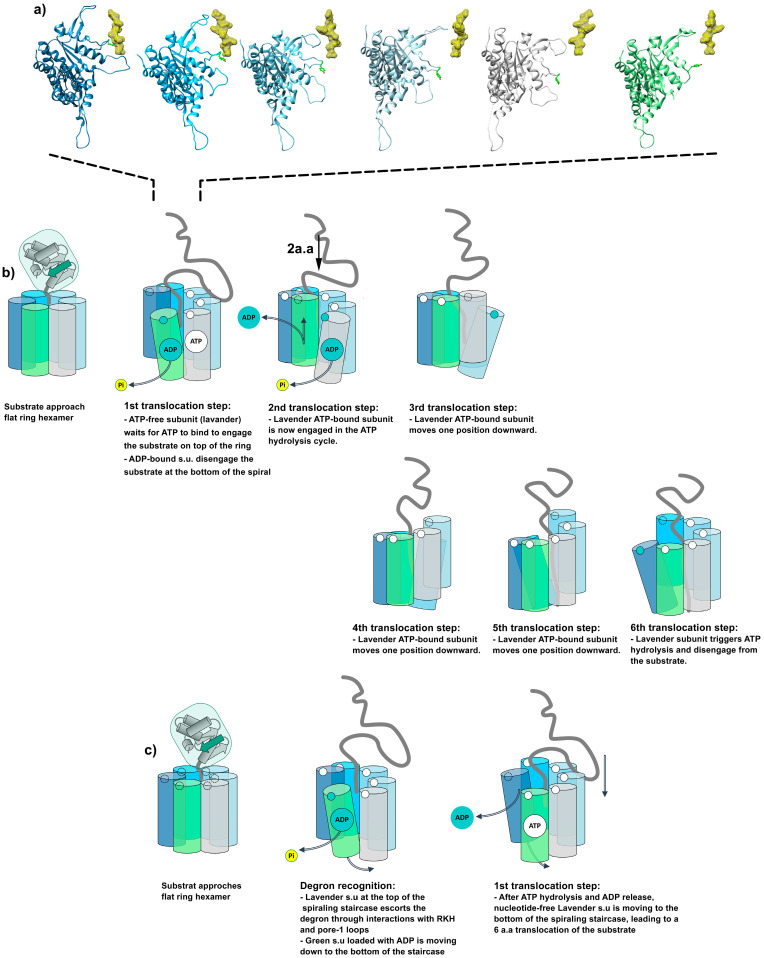
(**a**) Representation of the six ClpX protomers in interaction with the substrate, as observed in the cryo-EM structure of the substrate-engaged ClpX hexamer. The six protomers are represented with the same orientation as protomer #A in panel (**c**). The height of the substrate has been kept constant for the six protomers to highlight the differences in the positions of the axial loops from the six protomers. (**b**) Sequential model of substrate translocation triggered by a sequential ATP hydrolysis of the protomer located at the bottom of the helical staircase. Each hydrolysis event allows to pull the substrate by 2 amino acids. (**c**) Probabilistic model of substrate translocation, where the translocation can be triggered by any of the ClpX protomers. Upon ATP hydrolysis, the ATP-bound protomer position itself at the bottom of the staircase, thus pulling the substrate. The length of the translocation step then depends on the initial position of the protomer compared to the bottom of the staircase.

## Data Availability

No new data were created or analyzed in this study.

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
