# Peer review of "Recent Advances in the Structural Studies of the Proteolytic ClpP/ClpX Molecular Machine"

_biomolecules, 2025, doi:10.3390/biom15081097_

Round 1
Reviewer 1 Report
Comments and Suggestions for Authors
The paper “Recent Advances in the Structural Studies of the Proteolytic ClpP/ClpX Molecular Machine” by Audibert et al. is a review on the structures of ClpP/ClpX that try to explain some of the mechanisms of this complex molecular machine at a near atomic level.
Summarizing the results of such large and complex molecular machines is not simple, and the paper describes all the structural studies on Clp ClpX/ClpP by X-ray crystallography and cryo-EM, plus some side results obtained by NMR.
The paper is enriched by several figures that give an idea of the 3D movements of the different complexes and explain the mechanisms by using static pictures. At the end of the paper the presence of a movie (and Tables) is mentioned, but they are not available at the link.
In conclusion, a comprehensive and detailed review.
Author Response
|
Thank you very much for taking the time to review this manuscript, and the positive comments on our review. Please find the detailed responses below and the corresponding revisions/corrections highlighted/in track changes in the re-submitted files. |
|
Point-by-point response to Comments and Suggestions for Authors |
|
Comments 1: At the end of the paper the presence of a movie (and Tables) is mentioned, but they are not available at the link |
|
Response 1: Thank you for pointing this out. This review does not have supporting information. Therefore, we have removed the corresponding section at the end of the review (page 24). |
Reviewer 2 Report
Comments and Suggestions for Authors
The authors discussed on cryo-electron microscopy structural analysis of the AAA+ATPase (mainly ClpXP) to introduce the classic findings prior to high-resolution electron microscopy and review the process of arriving at a new model of the motor mechanics by structural analysis by electron microscopy with near-atomic resolution. Since AAA+ATPase is one of the central subjects of structural biology, with a large volume of literature, and was eagerly anticipated as a review for general molecular biologists.
The authors concisely summarize numerous literature sources and explain the concepts obtained in 15 figures. Even someone like me, who is not particularly familiar with this protein family, can understand it, so I recommend the manuscript to be published in the present form.
The biggest improvement I can suggest as a minor revision is the deficiency of discussion on the ratchet mechanism of this motor. Since peptides are inserted into the enzyme in one direction, there should be a mechanical part(s) reducing the probability of reversal motion in the opposite direction more than that of forward movement. In RNA polymerase, for example, it is shown in Bar-Nahum et al., Cell. 2005. doi: 10.1016 . Peptide hydrolysis and exclusion could be a candidate, but a clear presentation or discussion is expected.Author Response
|
Thank you very much for taking the time to review this manuscript, and the positive comments on our review. Please find the detailed responses below and the corresponding revisions/corrections highlighted in red or using track changes in the re-submitted files. |
|
Point-by-point response to Comments and Suggestions for Authors |
|
Comments 1: The biggest improvement I can suggest as a minor revision is the deficiency of discussion on the ratchet mechanism of this motor. Since peptides are inserted into the enzyme in one direction, there should be a mechanical part(s) reducing the probability of reversal motion in the opposite direction more than that of forward movement. In RNA polymerase, for example, it is shown in Bar-Nahum et al., Cell. 2005. doi: 10.1016 . Peptide hydrolysis and exclusion could be a candidate, but a clear presentation or discussion is expected. |
|
Response 1: Thank you for pointing this out. We have added a comment at the end of section 6.2 (p. 20, lines 657–661) in the revised version to address this point. “In the proposed mechanism, the hydrolysis of the ATP only occurs in the most basal subunit before ascending to the top of the helix. The helical structure of ClpX as a whole creates a ratchet mechanism. The displacement of the subunit is irreversible, preventing the system from moving backwards. Additionally, peptide hydrolysis and diffusion out of ClpP render substrat protein transfer from ClpX to ClpP irreversible.” |
Reviewer 3 Report
Comments and Suggestions for Authors
This manuscript is a review of structural studies on ClpXP, particularly recent CryoEM and NMR studies that provide insight into mechanism. The review is thorough and comprehensive, and provides a useful summary of the state of the field in terms of structure. It might be helpful if more of the biochemical studies that have also led to our current understanding of ClpXP mechanism (ie mutagenesis, kinetics, etc) were mentioned in the context of the structure – this was done for some single molecule experiments but otherwise was mostly absent.
Specific comments.
-Abstract. The final sentence in the abstract is “In addition, this review focuses on accessing additional dynamic information through the combined use of NMR with high resolution structural approaches to overcome bottlenecks towards understanding the molecular details that govern the protein degradation process by such molecular machines.” Although there was some discussion of NMR approaches, particularly with regard to studies of the 20S pore, I didn’t really see an emphasis on combining NMR with x-ray and cryo-EM throughout. Instead, it appears that the review serves more as a suggestion to the field that dynamic methyl-NMR techniques would serve to further our understanding of ClpXP catalysis.
-Figure 1: not sure what “evolutionary history of AAA+ ATPase” means (2004), or if it fits in the structural category. Similarly, not sure what “ClpXP ensure protein quality by degrading ssrA-tagged substrate” in 2007 is referring to – the ssrA tag has been known since far earlier than that, at least since a 1998 paper from the Sauer lab (Gottesman et al Genes & Dev). I would suggest the figure be revised, and also please include references along with each of the images on the timeline.
-Line 331: p97 isn’t a ribosomal protein, it’s an unfoldase with two sets of AAA+ rings
-Around line 471-472, sometimes subunits are referred to using hashtags (#A), sometimes without, unclear why.
-Line 520: 26S proteasome, not proteasome 26S
-Conclusion: 1st sentence of 2nd paragraph (line 669) seems disconnected from rest of that paragraph. Final paragraph indicates that a weakness of CryoEM is the lack of kinetic data. Perhaps the large amount of biochemical (albeit not structural) data on the ClpXP system should be acknowledged?
-Author contributions and funding not completed.
Comments on the Quality of English LanguageThere are grammatical issues throughout the manuscript. Most of the grammatical errors are minor, but some might make it difficult for a reader to follow.
Author Response
|
Thank you very much for taking the time to review this manuscript, and the positive comments on our review. Please find the detailed responses below and the corresponding revisions/corrections highlighted in red or using track changes in the re-submitted files. |
|
Point-by-point response to Comments and Suggestions for Authors |
|
Comments 1: It might be helpful if more of the biochemical studies that have also led to our current understanding of ClpXP mechanism (ie mutagenesis, kinetics, etc) were mentioned in the context of the structure – this was done for some single molecule experiments but otherwise was mostly absent.
Response 1: This review focuses on the structures of ClpP/ClpX to shed light on some of the mechanisms of this complex molecular machine at near-atomic levels. A huge body of literature and structural data exists on this topic. As acknowledged by other reviewers, our aim was to provide a comprehensive and detailed summary of this rapidly growing field. This review is already 28 pages long and illustrated with 15 carefully designed figures. While we recognise the importance of biochemical studies for understanding ClpXP functions and mechanisms, this is clearly a different topic which would require an independant review of a similar length. As the authors are not experts in ClpXP biochemical studies, we have chosen to focus on structural studies and leave the writing of a review dedicated to ClpXP biochemical studies to more specialised authors. As the reviewers mentioned, the structural part is already accompanied by the necessary functional details and their interrelations. We therefore do not wish to modify the structure of the paper and will leave it unchanged to maintain its fluidity, interest, comprehensibility, and readability.
Comments 2: Abstract. The final sentence in the abstract is “In addition, this review focuses on accessing additional dynamic information through the combined use of NMR with high resolution structural approaches to overcome bottlenecks towards understanding the molecular details that govern the protein degradation process by such molecular machines.” Although there was some discussion of NMR approaches, particularly with regard to studies of the 20S pore, I didn’t really see an emphasis on combining NMR with x-ray and cryo-EM throughout. Instead, it appears that the review serves more as a suggestion to the field that dynamic methyl-NMR techniques would serve to further our understanding of ClpXP catalysis. Response 2: Thank you for pointing this out. We agree with this comment. We have therefore modified the last sentence on page 1, lines 26–29, of the abstract: “In addition, this review focuses on accessing additional dynamic information through the combined use of NMR with high resolution structural approaches to overcome bottlenecks towards understanding the molecular details that govern the protein degradation process by such molecular machines” was replaced by : “In addition, this review presents some additional dynamic information obtained using solution-state NMR. This information provides molecular details that help to explain the protein degradation process by such molecular machines.
Comments 3: Figure 1: not sure what “evolutionary history of AAA+ ATPase” means (2004), or if it fits in the structural category. Similarly, not sure what “ClpXP ensure protein quality by degrading ssrA-tagged substrate” in 2007 is referring to – the ssrA tag has been known since far earlier than that, at least since a 1998 paper from the Sauer lab (Gottesman et al Genes & Dev). I would suggest the figure be revised, and also please include references along with each of the images on the timeline. Response 3: Thank you for pointing this out. We agree with the comment. We have modified Figure 1 to address these comments.
Comments 4: Line 331: p97 isn’t a ribosomal protein, it’s an unfoldase with two sets of AAA+ rings Response 4: Thank you for pointing this out. We have modified it in the text (line 377).
Comments 5: Around line 471-472, sometimes subunits are referred to using hashtags (#A), sometimes without, unclear why. Response 5: Thank you for pointing this out. We have modified it in the text (line 531).
Comments 6: Line 520: 26S proteasome, not proteasome 26S Response 6: Fixed (line 580).
Comments 7: Conclusion: 1st sentence of 2nd paragraph (line 669) seems disconnected from rest of that paragraph. Response 7: This sentence has been omitted from the revised version. (line 537).
Comments 8: Final paragraph indicates that a weakness of CryoEM is the lack of kinetic data. Perhaps the large amount of biochemical (albeit not structural) data on the ClpXP system should be acknowledged? Response 8: Thank you for pointing this out. We agree with this comment. We have now acknowledged the complementarity of biochemical data in the final sentence of the conclusion (lines 758–766): “In the future, applications of advanced solution state methyl-NMR approaches combined with biochemical studies are expected to provide complementary information to high-resolution cryo-EM structural data regarding the dynamic ballet and timing of each key steps in the functional cycle of the ClpXP machinery.”
Comments 9: Author contributions and funding not completed. Response 9: Thank you for pointing this out. We have completed these sections in the revised version (page 24).
Comments 10: There are grammatical issues throughout the manuscript. Most of the grammatical errors are minor, but some might make it difficult for a reader to follow. Response 10: Thank you for pointing this out. We have checked the text carefully and made several corrections to improve the grammar. The corrections are indicated in red or using the 'track changes' mode. |
Reviewer 4 Report
Comments and Suggestions for Authors
This review is devoted to the infinite and interesting topic of the functioning and structures of a large macromolecular machine ClpXP. This object is an AAA+ ATPase and is presented by complexes of two multimeric proteins consisting of six ClpX and twelve ClpP subunits. Of course, the structure of such a large and complicated complex is extremely difficult to determine. The authors convincingly and excitingly describe the history of its study from the first X-ray structures of the ClpXP components to modern high-resolution cryo-electron microscopy structure of the completely macromolecular machine. The structural part is accompanied by functional details and their interrelations. In the course of the narrative, the structures of the two components are described in detail. The story is supplemented by excellent illustrations and diagrams. As a result, the text is read with pleasure and becomes really interesting.
However, there are several small but important remarks regarding these illustrations and some sentences.
- Key words are missing.
- Figure 1. The timeline is formatted with errors. Please place the insertion in the correct place. Moreover, I believe that multiple references are not needed in the figure legend.
- Line 98. There should be a comma or a verb after ClpXP.
- Figure 2. Perhaps it should be noted here that the ssrA sequence is at the C-terminus of the polypeptide chain? This is important because below (line 135) it says that ClpX recognizes the unstructured N-terminal or C-terminal end of proteins, or this tag.
- Line 153. Here and everywhere, E. coli should be italicized.
- Figure 4. This is the figure I have the most criticism for. The positions of the amino acid side chains are unreadable. I can assume that Arg170, like Met98, has two alternative positions. Only one of them should be shown. This will not affect the essence of the contact diagram. The main chain of Ser97 and Met98 residues is shown poorly. The color scheme of protein atoms and hydrogen bonds is inconvenient and unreadable. Please remake this image. Your Figure 8 is a good example of how this can be done.
- Figure 5. Please place the full name of ADEP here, since this is where it appears for the first time.
Author Response
|
Thank you very much for taking the time to review this manuscript, and the positive comments on our review. Please find the detailed responses below and the corresponding revisions/corrections highlighted in red or using track changes in the re-submitted files. |
|
Point-by-point response to Comments and Suggestions for Authors |
|
Comment 1 : Key words are missing. Response 1 Thank you for pointing this out. We have completed this section in the revised version (page 1 – line 30).
Comment 2 : Figure 1. The timeline is formatted with errors. Please place the insertion in the correct place. Moreover, I believe that multiple references are not needed in the figure legend. Response 2: Thank you for pointing this out. We have checked the exact dates and now provide the references within the figure, as suggested by Reviewer 3.
Comment 3 : Line 98. There should be a comma or a verb after ClpXP. Response 3: Thank you for pointing this out. We have modified it in the text (line 103).
Comment 4 : Figure 2. Perhaps it should be noted here that the ssrA sequence is at the C-terminus of the polypeptide chain? This is important because below (line 135) it says that ClpX recognizes the unstructured N-terminal or C-terminal end of proteins, or this tag. Response 4: Thank you for pointing this out. We agree with your comment. In the revised version, the relevant information has been added to the legend of Figure 2 (lines 127-128).
Comment 5 : Line 153. Here and everywhere, E. coli should be italicized. Response 5: Thank you for pointing this out. In the revised version, E. coli is italicised throughout the text (shown in red or in tracking changes mode).
Comment 6 : Figure 4. This is the figure I have the most criticism for. The positions of the amino acid side chains are unreadable. I can assume that Arg170, like Met98, has two alternative positions. Only one of them should be shown. This will not affect the essence of the contact diagram. The main chain of Ser97 and Met98 residues is shown poorly. The color scheme of protein atoms and hydrogen bonds is inconvenient and unreadable. Please remake this image. Your Figure 8 is a good example of how this can be done. Response 6: Thank you for pointing this out. Following the comments, a new figure 4 is provided in this revised version.
Comment 7 : Figure 5. Please place the full name of ADEP here, since this is where it appears for the first time. Response 7: Thank you for pointing this out. The full name of ADEP (acyldepsipeptide) has been added in legend of figure 5 (line 231). |